# Direct cysteine sulfenylation drives activation of the Src kinase

David E. Heppner [1,5,6], Christopher M. Dustin[1], Chenyi Liao[2], Milena Hristova[1], Carmen Veith[1] Andrew C. Little[1], Bethany A. Ahlers[3], Sheryl L. White[4], Bin Deng[3], Ying-Wai Lam[3], Jianing Li [2] & Albert van der Vliet[1]

The Src kinase controls aspects of cell biology and its activity is regulated by intramolecular structural changes induced by protein interactions and tyrosine phosphorylation. Recent studies indicate that Src is additionally regulated by redox-dependent mechanisms, involving oxidative modification(s) of cysteines within the Src protein, although the nature and molecular-level impact of Src cysteine oxidation are unknown. Using a combination of biochemical and cell-based studies, we establish the critical importance of two Src cysteine residues, Cys-185 and Cys-277, as targets for $H_2O_2$-mediated sulfenylation (Cys-SOH) in redox-dependent kinase activation in response to NADPH oxidase-dependent signaling. Molecular dynamics and metadynamics simulations reveal the structural impact of sulfenylation of these cysteines, indicating that Cys-277-SOH enables solvent exposure of Tyr-416 to promote its (auto)phosphorylation, and that Cys-185-SOH destabilizes pTyr-527 binding to the SH2 domain. These redox-dependent Src activation mechanisms offer opportunities for development of Src-selective inhibitors in treatment of diseases where Src is aberrantly activated.

[1] Department of Pathology and Laboratory Medicine, Robert Larner, M.D. College of Medicine University of Vermont, 149 Beaumont Avenue, Burlington, VT 05405, USA. [2] Department of Chemistry, College of Arts and Sciences, University of Vermont, 82 University Place, Burlington, VT 05405, USA. [3] Department of Biology, College of Arts and Sciences, University of Vermont, 109 Carrigan Drive, Burlington, VT 05405, USA. [4] Department of Neurological Sciences, Robert Larner, M.D. College of Medicine University of Vermont, 149 Beaumont Avenue, Burlington, VT 05405, USA. [5] Present address: Department of Cancer Biology, Dana-Farber Cancer Institute, 450 Brookline Ave, Boston, MA 02215, USA. [6] Present address: Department of Biological Chemistry and Molecular Pharmacology, Harvard Medical School, 240 Longwood Ave, Boston, MA 02115, USA. These authors contributed equally: David E. Heppner, Christopher M. Dustin, Chenyi Liao. Correspondence and requests for materials should be addressed to D.E.H. (email: david_heppner@dfci.harvard.edu) or to J.L. (email: jianing.li@uvm.edu) or to A.V. (email: albert.van-der-vliet@med.uvm.edu)

The proto-oncogene protein tyrosine kinase Src is the prototypical member of the Src-family kinases (SFKs) that participate in cell signaling pathways by catalyzing phosphorylation of specific tyrosine residues in various target proteins[1]. Commonly activated by initial activation of cell surface receptors, Src controls various cellular outcomes, including differentiation, adhesion, migration, and proliferation[2,3]. As the first characterized proto-oncogene, it is well appreciated that aberrant Src activation and expression is associated with malignant transformation and oncogenesis[4] establishing Src as a chemotherapeutic target in the treatment of various cancers[5]. Additionally, pharmacological inhibition of Src and other SFKs have been shown to be effective in several nonmalignant human diseases[6,7]. Therefore, elucidation of factors that regulate Src activation is critical to understanding its extensive roles in human disease and for development of effective treatments.

A nonreceptor tyrosine kinase, Src activity is regulated through protein structural changes triggered by intramolecular domain interactions through Src homology (SH) 2 and 3 domains and by (de)phosphorylation of key tyrosine residues, thereby coupling activation of Src with targeting of appropriate cellular substrates[8]. In its autoinhibited form, Src is phosphorylated at Tyr-527 (pTyr) (chicken sequence numbers used throughout) within the C-terminal tail, which promotes its binding to the SH2 domain, maintaining the protein in a minimally active "clamped" confirmation[9]. Upon dephosphorylation of pTyr-527, Src unfolds inducing several structural changes, which allows for binding to downstream targets[9]. The structural hallmark of the maximally active Src kinase is the unfolded activation loop (A-loop) α-helix, which exposes Tyr-416 for phosphorylation and sustains maximal kinase activity[9]. Molecular modeling studies describe a dynamic molecular model for Src kinase activation involving initial conversion of the autoinhibited kinase to an active-like state, in a two-step process with A-loop unfolding followed by αC-helix rotation[10,11]. These two states exist in equilibrium, favoring the autoinhibited conformation. Subsequent (auto)phosphorylation of Tyr-416 by intermolecular encounter with another active kinase then stabilizes the active form of Src[11,12]. In addition to Tyr-416, phosphorylation of additional tyrosines may also regulate SFK function[13].

In addition to regulation by tyrosine (de)phosphorylation, accumulating evidence indicates that Src activation occurs in association with increased cellular production of reactive oxygen species (ROS)[14]. ROS generated from NADPH oxidases (NOX), respiring mitochondria, or other sources are capable of modulating signaling pathways by reversible oxidation of conserved cysteine (Cys-SH) residues within target proteins[15,16]. Such reversible redox modifications have been implicated in regulation of tyrosine phosphorylation, which is largely attributed to inactivation of protein tyrosine phosphatases by reversible oxidation of their catalytic cysteines, thereby resulting in enhanced or extended tyrosine phosphorylation[17]. However, tyrosine kinases themselves are also subject to direct redox regulation by oxidation of noncatalytic cysteines[18–20]. Indeed, tyrosine kinases such as the epidermal growth factor receptor (EGFR) and SFKs interact directly with NOX enzymes during their activation[21–23], and recent studies by our group[22,24–26] and others[21,27] indicate that NOX-mediated activation of Src and EGFR closely associates with cysteine oxidation within these kinases. The Src protein contains nine cysteine residues, most of which are conserved among SFKs and related kinases (Supplementary Fig. 1 and Supplementary Table 1), and studies with cysteine mutants have suggested the involvement of several of these cysteines in ROS-mediated Src activation[28–32]. However, the molecular mechanisms by which cysteine oxidation promotes Src kinase activity remain unclear, and studies with recombinant Src proteins confoundingly indicate that ROS or other thiol-reactive agents can also inactivate kinase activity[28,30,31].

Oxidation of cysteine by $H_2O_2$, the main mediator of NOX-mediated redox signaling, initially generates a sulfenic acid (Cys-SOH), but generates additional oxidative modifications in subsequent reactions[25]. Recent studies indicate that redox-dependent activation of tyrosine kinases such as Src or EGFR closely associates with initial formation of Cys-SOH, the proximal product of cysteine oxidation by $H_2O_2$, and that other oxidative modifications such as S-glutathionylation do not enhance kinase activity[21,25,33,34]. The present studies were conducted to characterize the redox-sensitive cysteine residues within Src that are most susceptible to sulfenylation, and to identify the impact of this modification on tyrosine kinase activity. We identified Cys-185 and Cys-277 as the primary Cys residues oxidized to Cys-SOH by $H_2O_2$, and demonstrated that oxidation of both cysteines is critical for redox-dependent kinase activation. Molecular dynamics (MD) and metadynamics simulations indicate that modification of these cysteines to Cys-SOH induces local structural changes that directly impact the regulatory pTyr-527 and Tyr-416 residues, respectively.

## Results

**Sulfenylation of Cys-185 and Cys-277 enhances Src activity.** To identify the redox-active Src cysteine residues that are susceptible to sulfenylation by $H_2O_2$, we utilized various sulfenic acid-selective trapping reagents to assess $H_2O_2$-induced cysteine modifications in recombinant Src using liquid chromatography-tandem mass spectrometry (LC–MS/MS). First, recombinant Src (~20 µg) was treated with various concentrations of $H_2O_2$ (0.1–100 mM) in the presence of 5,5′-dimethyl-1,3-cyclohexanedione (dimedone) to trap Cys-SOH species, and then trypsin digested for MS analysis. These analyses typically yielded ~80% peptide coverage, comprising all nine cysteine residues. In the absence of $H_2O_2$, all cysteines were detected as iodoacetamide adducts, and no dimedone adducts were observed (except at Cys-498), suggesting they were reduced. In contrast, upon $H_2O_2$ treatment, Cys-S-dimedone modifications were observed at three cysteines (Cys-185, Cys-277, and Cys-498), indicating that they were sulfenylated by $H_2O_2$ and trapped as dimedone adducts (Supplementary Fig. 2; Supplementary Table 2 and Supplementary Data 1). Other cysteine modifications, such as sulfinic and sulfonic acids, were also identified on several cysteines, but were also observed in untreated Src (Supplementary Data 1), presumably due to autoxidation during purification and processing. Oxidation of Src by high doses of $H_2O_2$ (100 mM) revealed overoxidized Cys-185 and Cys-277 (to sulfonic acids; Supplementary Data 1), which further supports the notion that Cys-185 and Cys-277 are redox active. Importantly, while no Cys-S-dimedone adducts were observed on other cysteines, oxidation of Src with high amounts of $H_2O_2$ (>1 mM) induced DTT-reducible mass shifts upon analysis by Coomassie stained sodium dodecyl sulfate polyacrylamide gel electrophoresis (SDS-PAGE), indicative of intra- and/or intermolecular disulfide crosslinking, which were in some cases independent of Cys-185 or Cys-277 (Supplementary Fig 3).

One limitation of dimedone is its relatively slow reaction rate with sulfenic acids, suggesting that some intermediate Cys-SOH may have gone undetected. We, therefore, performed similar analysis with an alternative sulfenic acid-reactive probe, bicyclo [6.1.0]nonyne (BCN), which is known to react with Cys-SOH with a 100-fold faster rate to form –S(O)-BCN adducts[35,36]. Global LC–MS/MS analysis did not reveal formation of –S(O)-BCN adducts in unreacted Src, but adducts were detected on three cysteines during reaction of Src with $H_2O_2$, namely

**Table 1 Quantitative analysis of Src sulfenylation using LC-MS.**

| Cysteine Residue | Peptide Sequence | Charge | m/z | H:L ratio[a] | SD[b] |
|---|---|---|---|---|---|
| Cys-185 | GAYC(dim)LSVSDFDNAK | 2+ | 814.369 | 15.48 | 2.25 |
|  | GAYC(dim-d6)LSVSDFDNAK | 2+ | 817.388 |  |  |
| Cys-238 | HADGLC(dim)HR | 2+ | 523.745 | 17.36 | 11.44 |
|  | HADGLC(dim-d6)HR | 2+ | 526.764 |  |  |
| Cys-245 | LTTVC(dim)PTSKPQTQGLAK | 3+ | 637.680 | 2.25 | 1.40 |
|  | LTTVC(dim-d6)PTSKPQTQGLAK | 3+ | 639.693 |  |  |
| Cys-277 | LGQGC(dim)FGEVWMGTWNGTTR | 2+ | 1119.509 | 78.57 | 5.54 |
|  | LGQGC(dim-d6)FGEVWMGTWNGTTR | 2+ | 1122.528 |  |  |
| Cys-400 | AANILVGENLVC(dim)KVADFGLAR | 2+ | 741.405 | 13.07 | 9.84 |
|  | AANILVGENLVC(dim-d6)KVADFGLAR | 2+ | 744.424 |  |  |
| Cys-483 | MPC(dim)PPECPESLHDLMCQCWR [c] | 2+ | 1342.548 | 31.39 | 24.08 |
|  | MPC(dim-d6)PPECPESLHDLMCQCWR [c] | 2+ | 1345.567 |  |  |

All dimedone/dimedone-d6 labeled Src peptides were identified using targeted mass spectrometry analyses. Five out of six dimedone/dimedone-d6 labeled Src peptide pairs were quantified within an average mass error of less than 5 ppm across the peak of elution. XCalibur was used to quantify MPC(dim/dim-d$_6$)PPECPESLHDLMCQCWR using m/z 1342.548 → [y3]$^+$, [y7]$^+$, and [y13]$^+$ (dim labeled) and m/z 1345.567 → [y3]$^+$, [y7]$^+$, and [y13]$^+$ (dim-d$_6$ labeled)
[a]Heavy to light ratio
[b]Standard deviation ($n = 3$)
[c]Quantification of Cys-483 H/L ratio was performed in Xcalibur using three transitions. The three cysteines c-terminal to Cys-483 are carbamidomethylated. All other peptides were quantified by Skyline using four to six transitions from the targeted mass spectrometry experiments with PRM

Cys-185, Cys-245, Cys-277, and not observed on any other cysteines (Supplementary Fig. 4, Supplementary Table 2, and Supplementary Data 2). Importantly, BCN can also react with Cys-SH[37,38], but this caveat is easily addressed since reaction with Cys-SOH yields a peptide [Cys-S(O)-BCN] + 16 Da heavier than reaction products with a thiol (Cys-S-BCN). Indeed, our MS analysis did not reveal detectable peptides containing Cys-S-BCN in unreacted or $H_2O_2$-treated Src. Finally, we also assessed cysteine sulfenylation in the presence of 4-chloro-7-nitrobenzofurazan (NBD-Cl)[33,39] and observed that $H_2O_2$-treated Src contained detectable Cys-S(O)-NBD adducts at Cys-277 and Cys-498, consistent with Cys-SOH formation at these residues (Supplementary Table 2 and Supplementary Data 3).

To more rigorously identify and quantify the most susceptible redox-sensitive Src Cys residues, we performed quantitative MS analysis of dimedone trapping using labeling with light dimedone (dim-d$_0$) and heavy dimedone (dim-d$_6$)[40] of nonoxidized and $H_2O_2$-treated Src, respectively. Targeted mass spectrometry with parallel reaction monitoring (PRM) was performed to monitor all cysteine-containing peptides with dimedone/dimedone-d$_6$ modifications. The targeted analysis not only further confirmed the dimedone labeled peptides identified from the exploratory mass spectrometry experiments with higher S/N and sequence coverage, but also identified other low-abundant dimedone containing peptides that were not identified in the previous mass spectrometry runs (Table 1 and Supplementary Data 4). All the measured precursor masses of the identified Src peptides were within 0.5 ppm of the theoretical masses and the tandem mass spectra (MS/MS) exhibited a continuous stretch of y-ion series, and clear peak assignments, indicating confident identifications (Supplementary Data 5). These analyses revealed Cys-277 and Cys-185 as the most prominently oxidized cysteines, consistent with previous trapping experiments. In case of Cys-277, distinct peaks were observed for dim-d$_6$ and dim-d$_0$ peptide ion isotopologues in the MS1 spectrum (Fig. 1a). Extracted chromatograms for dim-d$_6$ and dim-d$_0$ labeled ions (Fig. 1b red and black, respectively) enabled the determination of heavy to light (H/L) dimedone tagging ratios indicating the degree of cysteine oxidation compared to nonoxidized controls. MS/MS fragmentation spectra of dim-d$_6$ and dim-d$_0$ dimedone labeled Cys-277-containing peptide ions confirmed the identity of this peptide with expected Cys-S-dim-d$_{0/6}$ isotope-dependent mass shifts (Supplementary Fig. 5C, D). Similar analysis was performed

for Cys-185 (Supplementary Fig. 5E–H) and the other cysteine-containing peptides (Table 1 and Supplementary Data 4). Although these analyses also suggested increased sulfenylation of other cysteines, as indicated by H/L ratios >1, these observations were considered less robust based on the high variance (standard deviation (SD)) between triplicate replicates (Table 1 and Supplementary Data 4). Collectively, these analyses indicate that Cys-185 and Cys-277 were consistently identified as the main redox-sensitive cysteines within Src that form detectable Cys-SOH species by $H_2O_2$. Previous reports have implicated Cys-277[30,32] and other cysteines (e.g., Cys-245[29] and Cys-498[28,31]) as redox-sensitive cysteines in the context of ROS signaling, while Cys-185 has yet to be identified in such mechanisms. The structure of autoinhibited Src (PDB ID: 2SRC) reveals that Cys-185 and Cys-277 reside near (~5–10 Å) pTyr-527 and Tyr-416, respectively, possibly indicating that their oxidation may directly impact these critical regulatory tyrosines (Supplementary Figs. 1B and 6)[9]. In contrast, Cys-245 and Cys-498 are distant from these tyrosines (~20–30 Å) and located in regions that are not significantly altered during Src activation. From these considerations, our subsequent studies focused on the role of Cys-185 and Cys-277 oxidation in regulating Src activation.

We evaluated tyrosine kinase activity of recombinant Src in the presence of $H_2O_2$ and, consistent with previous studies[25,41], observed that kinase activity of wild-type Src is enhanced linearly and correlated with $H_2O_2$ in a dose-dependent manner, ranging between 10 and 30 μM (Fig. 2b and Supplementary Fig. 7), although higher $H_2O_2$ concentrations (>100 μM) did not enhance and even inhibited activity (Supplementary Fig. 8). In contrast, $H_2O_2$ was incapable of statistically increasing activity of C185A or C277A Src variants, while both retained significant kinase activity. It is important to note that, while the absolute $H_2O_2$ concentrations in these studies were lower than those in the MS analyses, the relative molar excess of $H_2O_2$ over Src was considerably greater (necessary to overcome the high concentrations of DTT). In any case, these results indicate that $H_2O_2$ can directly enhance intrinsic Src tyrosine kinase activity, which is dependent on both Cys-185 and Cys-277, and consistent with these residues being the most susceptible to sulfenylation by $H_2O_2$.

**Cys-185 and Cys-277 are critical for Src activation in cells.** We next addressed the impact of Cys-185 and Cys-277 on

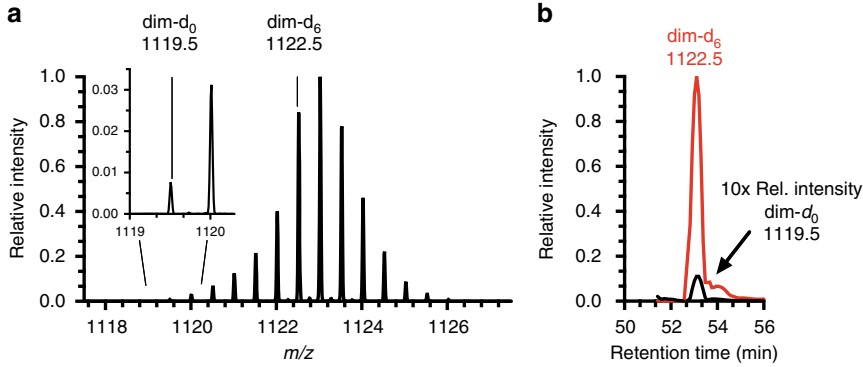

**Fig. 1** LC–MS/MS Quantification of Cys-277 sulfenylation. Recombinant Src ($\sim$0.6 µg/µL) was either incubated with 1.0 mM of light dimedone (dim-$d_0$) in the absence of $H_2O_2$ or treated with 1.0 mM $H_2O_2$ in the presence of 1.0 mM heavy dimedone (dim-$d_6$) to trap sulfenylated cysteine residues. Equal amounts of dim-$d_0$ and dim-$d_6$ Src were combined prior to trypsin digestion and LC–MS/MS analysis. **a** Representative MS1 spectrum exhibiting peaks corresponding to the dim-$d_0$- ($m/z$ 1119.5, inset) and dim-$d_6$-conjugated ($m/z$ 1122.5) Cys-277 containing peptide fragment LGQGCFGEVWMGTWNGTTR (2+ charge). **b** Representative extracted ion chromatograms (with deuterated version eluting slightly earlier) from peptide-targeted parallel reaction monitoring (PRM) of dim-$d_0$ (black, relative intensity multiplied by ten) and dim-$d_6$ (red) enabling the determination of relative amounts of dim-$d_6$/dim-$d_0$ corresponding to a heavy to light (H/L) ratio of 78.57 ± 5.54 (triplicate analysis). H/L ratio was calculated from the five transitions from three experimental replicates

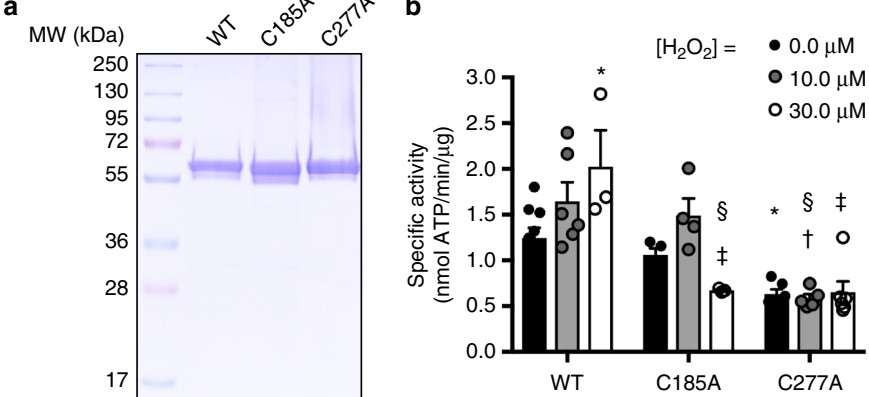

**Fig. 2** $H_2O_2$ enhances Src kinase activity depending on Cys-185 and Cys-277. **a** Coomassie-stained SDS-PAGE gel of 5 ng of WT, C185A, and C277A recombinant Src. **b** Analysis of kinase activity of recombinant Src variants following $H_2O_2$ treatment measured with ADP Glo kinase activity kit. Data are expressed as means ± s.e.m. of at least three replicates from three independent experiments. Analyzed with two-way ANOVA where [*]$p < 0.01$ compared to WT [$H_2O_2$] = 0 µM; [‡]$p < 0.0001$ compared to WT [$H_2O_2$] = 30 µM; [†]$p < 0.0001$ compared to WT [$H_2O_2$] = 10 µM; [§]$p < 0.005$ compared to C185A [$H_2O_2$] = 10 µM

$H_2O_2$-mediated Src activation in epithelial cells. H292 mucoepidermoid epithelial cells were either transiently or stably transfected with C-terminal FLAG-tagged Src (Src-FLAG) or Cys mutant Src constructs, and activation of Src-FLAG was assessed in response to stimulation with ATP to enhance cellular $H_2O_2$ production by activation of the NADPH oxidase DUOX1[22,25,42], by monitoring Src (auto)phosphorylation at Tyr-416. While these cells express endogenous Src, we confirmed that expressed Src-FLAG protein is distributed similarly with the cells (Supplemental Fig. 9), and is efficiently purified with M2 anti-FLAG methodologies (Supplemental Fig. 10). As expected[22,25], ATP stimulation of H292 cells expressing WT Src-FLAG resulted in its enhanced phosphorylation at Tyr-416 (Fig. 3, Supplementary Figs. 11–13). No significant changes were observed in Src phosphorylation at pTyr-527 (Supplementary Fig. 13), suggesting that ATP-mediated activation of Src might occur independent of pTyr-527 dephosphorylation. Comparative ATP stimulation of H292 cells expressing either C185A or C277A Src-FLAG variants showed reduced Tyr-416 phosphorylation within these proteins

(Fig. 3 and Supplementary Figs. 11, 13), indicating that these proteins were not activated in response to cellular $H_2O_2$ production. To associate these observations with cysteine oxidation, cells expressing WT, C185A, and C277A Src-FLAG were pre-loaded with the alkyne derivative of dimedone (DYn-2)[21] prior to stimulation. Following lysis, DYn-2-tagged proteins were conjugated to biotin-azide by click-mediated ligation and purified with M2 anti-FLAG antibodies, to visualize formation of Cys-SOH within Src-FLAG variants by streptavidin-horseradish peroxidase (HRP) blotting. Consistent with previous studies[25], ATP stimulation induced sulfenylation within WT Src, whereas sulfenylation was dramatically attenuated in C185A and C277A variants (Fig. 3 and Supplementary Figs. 11, 12), indicating that both Cys-185 and Cys-277 are sulfenylated during DUOX1-dependent activation of Src. Alternatively, ATP-stimulated H292 cells expressing Src-FLAG proteins were derivatized during cell lysis with biotin-conjugated dimedone (DCP-Bio1), after which Src-FLAG proteins were purified and analyzed for pTyr-416 or biotin incorporation, which yielded highly similar results

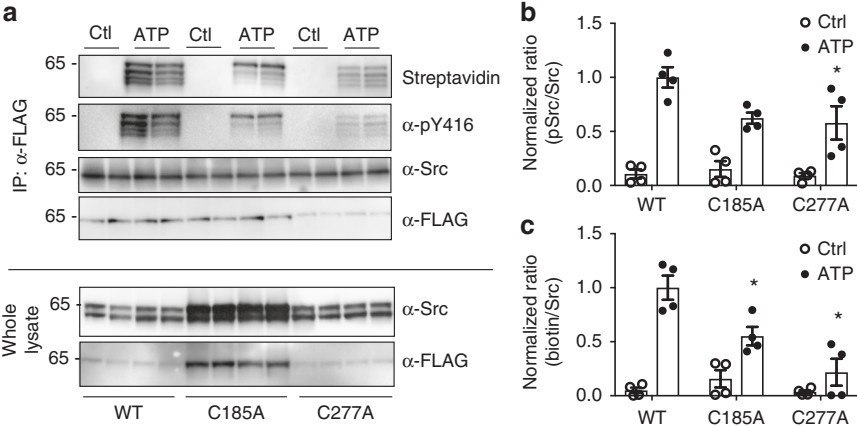

**Fig. 3** ATP induces Src sulfenylation at Cys-185 and Cys-277 that promote pTyr-416 phosphorylation in H292 cells. **a** H292 cells were transiently transfected with c-terminal FLAG-tagged WT, C185A, and C277A Src (Src-FLAG), and preloaded with DYn-2 and stimulated with 100 μM ATP for 10 min. Derivatized cell lysates or α-FLAG-purified were analyzed by western blot with indicated antibodies. **a** Representative western blots of two separate experiments. **b** Normalized ratios of phosphorylation-to-total Src. **c** Normalized ratios of biotinylation-to-total Src. Data represent mean ± s.e.m from at least four replicates. Data were analyzed with student $t$ test comparing control to ATP-treated cells. $^*p < 0.01$

(Supplementary Fig. 13). These collective findings indicate that Src is redox regulated within a cellular context and that both Cys-185 and Cys-277 are sulfenylated in response to DUOX1-mediated $H_2O_2$ production and are critical for Src activation.

**MD simulations reveal structural alterations by sulfenylation.** Activation of Src involves significant structural alterations, but it is unclear how sulfenylation of specific cysteines impact on kinase activation. To address this, we performed both microsecond-long MD simulations and metadynamics simulations to assess the structural perturbations induced by substitution of Cys-185 and Cys-277 in the autoinhibited Src structure with their sulfenylated form, Cys-SOH. It is not intuitive how such a small modification would induce significant structural alterations, but recent MD simulations indicate significant impact of sulfenylation of specific cysteines in other proteins[27,43]. As Cys-SOH is a weak acid, it may also be deprotonated (Cys-SO⁻). Various Cys-SOH $pK_a$ estimates indicate values ranging from 6 to 10 in the context of small molecules[44,45], consistent with density-functional theory calculations of model compounds (see Supplementary computational methods). Recent estimates in dipeptides and selected proteins indicate $pK_a$ values for Cys-SOH of ~5.9–7.2[38,46], implying that a significant fraction of Cys-SOH exist in its protonated state under physiological conditions. Based on this, and the fact that previous MD simulations of Cys-SOH indicated structural changes when modeling the protonated state[27,43], we performed our MD simulations based on protonated forms of Cys-SOH as well as Cys-SH. To incorporate the Cys-SOH into these MD simulations, we derived and benchmarked CHARMM36-compatible force field parameters for the cysteine sulfenic acid (cysteine sulfenic acid force field parameters in Supplementary Methods and Supplementary Fig. 14).

We first simulated the effect of Cys-SOH at Cys-277, which is located on the solvent-accessible G-rich loop and within ~10 Å from Tyr-416 on the A-loop α-helix[9]. The wild-type inactive Src structure (PDB ID: 2SRC), and other systems in which Cys-277 is reduced (Cys-SH), were considered control structures for comparing structural changes induced by Cys-277-SOH. The A-loop (residues 406–423), which consists of two helical segments (residues 407–410 and 414–418), remains folded throughout the MD simulations of all control structures (Supplementary Fig. 15; Supplementary Movie 1). In contrast,

when Cys-277-SH was substituted with Cys-277-SOH, a significant loss in A-loop helicity was observed in both simulations of 5 μs (Fig. 4b). One of these simulations showed rapid unfolding of the A-loop (#1 in Fig. 4b; Supplementary Movie 2), while the other underwent a slower process with major unfolding in the segment of res. 407–410 (#2 in Fig. 4b). These findings suggest that Cys-277-SOH-dependent A-loop unfolding is complex and involves multiple paths. Focusing on the interactions around the A-loop, we analyzed these two Cys-277-SOH simulations and the other six control simulations (all Cys-SH, Cys-185-SOH, and Cys-498-SOH). In control groups, the key regulatory Tyr-416 resides near or within H-bonding distance with Asp-386 of loop residues 383–389, with the side-chain phenol positioned away from solvent and within the catalytic cleft (Supplementary Fig. 15B). Additionally, these simulations identified another possible key residue, Arg-419, which occasionally swings to interact with Asp-386. Cys-277-SOH likely competes with Tyr-416 for H-bonding interactions with Asp-386 and a result of sulfenylation of Cys-277, interaction of Cys-277-SOH with Asp-386 promotes a shuffle in these residues resulting in pairing of Arg-385 with Asp-444 due to the displacement of loop residues 383–389 (Supplementary Figs. 15A and 16B). Subsequently, Arg-419 is able to pair with Asp-386, enabling the exposure of Tyr-416 to solvent making it accessible for phosphorylation.

The impact of Cys-277-SOH on A-loop unfolding is further evident from the free-energy landscapes obtained from metadynamics simulations (Fig. 4c). According to the interactions revealed from the above MD simulations, we chose the pair distances of (Tyr-416)–(Asp-386) and (Arg-419)–(Asp-386) as collective variables (CVs) and performed metadynamics simulations on the Cys-277-SOH and wild-type (Cys-277-SH) systems, respectively. Large increases of these pair distances due to sulfenylation of Cys-277 would be expected to lead to unfolding of the A-loop, thus further confirming the impact of Cys-277-SOH on A-loop unfolding. In the wild-type structure, the lowest free-energy conformation is found when Tyr-416 is ~5 Å from Asp-386 and Arg-419 is ~15 Å from Asp-386. However, when Cys-277 is replaced with Cys-SOH, the lowest free-energy conformations appear in two areas: a small region that is similar to the wild-type and a larger region where both Tyr-416 and Arg-419 are at significantly longer distances of 17–22 Å from Asp-386. This latter "long-distance" conformation corresponds to the

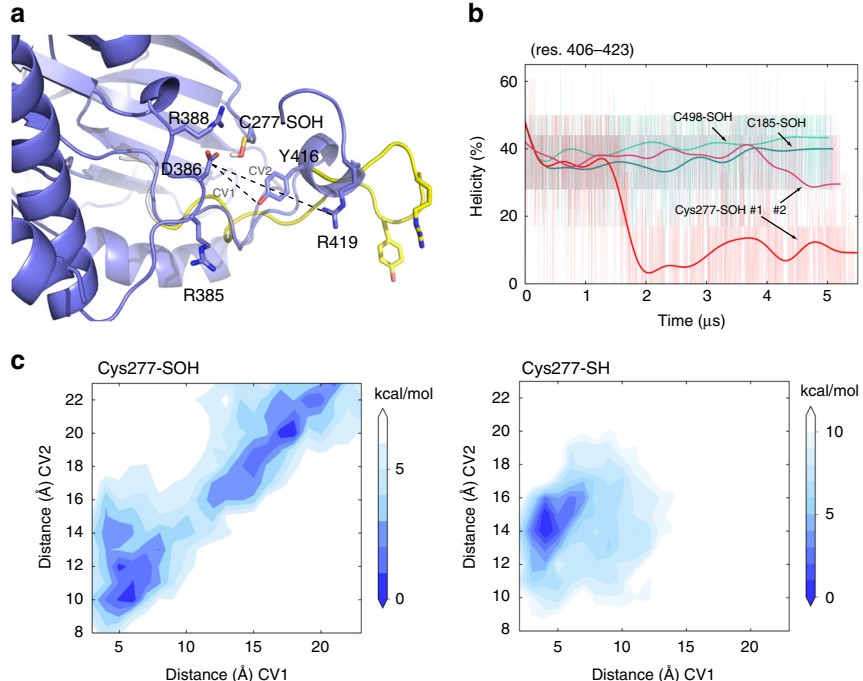

**Fig. 4** Cysteine oxidation to Cys-277-SOH promotes unfolding of the Src A-loop α-helix. **a** A final snapshot of Cys-277-SOH system (purple blue) from MD simulation superposed with it from metadynamics simulation that displays the unfolded conformation of the A-loop (residues 406–423, yellow). Detailed structures compared with wild-type group are shown in Supplementary Fig. 12. Key residues that interact with the A-loop region are labeled. **b** Percent α-helix fold of residues 406–423 throughout the MD simulation of Cys-277-SOH system (two replicas labeled as #1 and #2 are in red and pink) in comparison with control groups (green and cyan). **c** Free-energy landscapes of Cys-277-SOH system and Cys-277-SH system, respectively. Two collective variables (CVs), pair distances of Tyr-416-Asp-386 and Arg-419-Asp-386, are used in metadynamics simulations. Color scale: Cys277-SOH white = 7.0 kcal/mol, blue = 0.0 kcal/mol (left) and Cys277-SH white = 11.0 kcal/mol, blue = 0.0 kcal/mol

unfolded A-loop structure (yellow in Fig. 4a), which is energetically favorable. Results from these simulations, therefore, indicate that Cys-277-SOH induces activation of the kinase by promoting rapid and thermodynamically favorable unfolding of the A-loop and solvent exposure of Y416 for phosphorylation.

Next, we simulated the effect of sulfenylation of Cys-185, located within the SH2 domain near the regulatory active site that binds pTyr-527 located on the Src C-terminal tail. Simulations of reduced Cys-185-SH show that pTyr-527 forms H-bonds with Arg-175 and Arg-155, while Cys-185-SH, at a distance of ~4–6 Å away from pTyr-527 or Arg-175, is otherwise noninteracting with these residues (Supplementary Fig. 16A, Supplementary Movie 3). In contrast, when Cys-185-SH is substituted for Cys-185-SOH, it now engages in H-bonding with the phosphate moiety of pTyr-527 (Fig. 5a and Supplementary Movie 4). We did not observe full dissociation of pTyr-527 within the limited length of our simulations, but dissociation may be indicated by the increasing number of water solvent contacts with pTyr-527 induced by sulfenylation of Cys-185 as compared with reduced (Cys-185-SH) states (Fig. 5b), and such increased solvent exposure of pTyr-527 may enhance access to tyrosine phosphatases that promote its dephosphorylation and Src unclamping.

In line with these conventional simulations, metadynamics simulations revealed that Cys-185-SOH significantly alters the free-energy landscape of pTyr-527 binding to the SH2 domain. In these simulations, we chose pair distances of (pTyr-527)–(Arg-175) and (pTyr-527)–(Arg-155) as CVs, since these positive arginine side chains enable tight electrostatic binding of the negative pTyr-527 and observed increases in distances would model the dissociation of pTyr-527 from the SH2 domain. In wild-type Src (Cys-185-SH), the lowest free-energy conformation is observed when pTyr-527 is located nearby (~4–6 Å) Arg-175

and Arg-155 (Fig. 5c right), consistent with the structure of autoinhibited Src. In contrast, when Cys-185 is present as a sulfenic acid, the lowest free-energy conformation is located where pTyr-527 is completely dissociated from the SH2 domain, at a distance of 13–18 Å from Arg-175 and Arg-155 (yellow in Fig. 5a and Fig. 5c left). These findings further support the notion that oxidation of Cys-185-SH to Cys-185-SOH critically destabilizes the binding of the autoinhibitory pY527 in a mechanism where first solvent accessibility to pTyr-527 is enhanced via hydrogen bonding eventually leading to complete exergonic dissociation from the SH2 domain.

Finally, since our mass spectrometry analysis also indicated sulfenylation at Cys-498 (Supplementary Table 2), we also carried out simulations of two replicas of Cys-498-SOH for a total of 10 μs. However, only subtle conformational changes were observed at the helical bundles near Cys-498 (Supplementary Fig. 17A, B), and metadynamics simulations show comparable energy basins for the Cys-498-SH and Cys-498-SOH systems (Supplementary Fig. 17C). These findings together indicate minimal structural impact of Cys-498-SOH within the C-lobe, and thus a minimal direct role for Cys-498 oxidation for Src kinase activation.

## Discussion

While several studies have suggested that the Src kinase is regulated by direct redox mechanisms, the identity of the redox-sensitive cysteine residues and the mechanism(s) by which their oxidation impacts kinase activation are unknown[29,30]. Following our recent findings indicating that redox-dependent activation of Src in biochemical and cellular contexts closely associates with formation of sulfenic acids (Cys-SOH) within Src[24,25], we now identify Cys-185 and Cys-277 as the main cysteines oxidized to

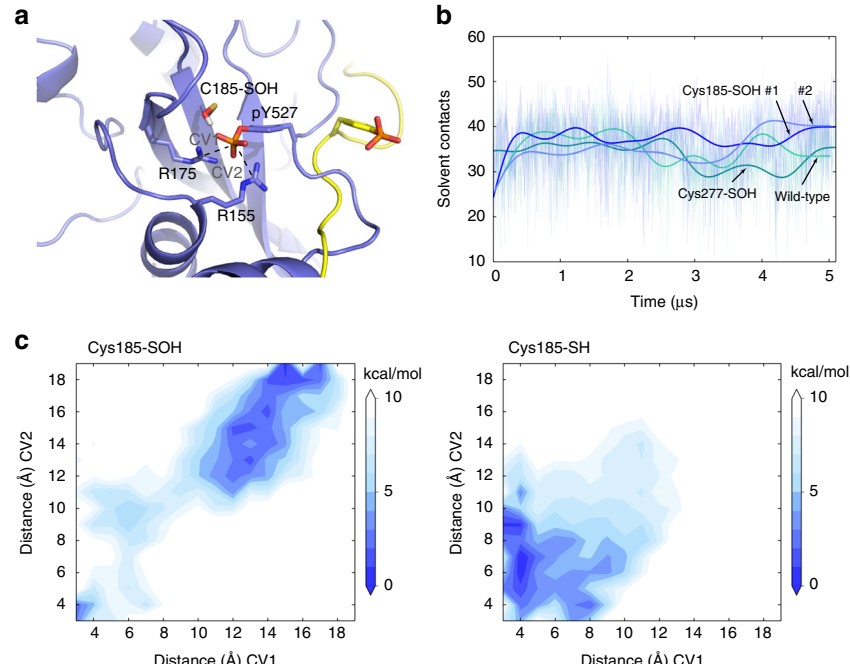

**Fig. 5** Sulfenylation of Cys-185 destabilizes autoinhibitory pTyr-527 binding to SH2 domain. **a** A final snapshot of Cys-185-SOH system (purple blue) from MD simulation superposed with it from metadynamics simulation that displays the dissociated pTyr-527 (yellow). Key residues that interact with pTyr-527 are labeled. **b** Number of water solvent contacts with pTyr-527 over time in Cys-185-SOH system (two replicas labeled as #1 and #2 are in blue and lightblue) in comparison with control groups (green and cyan). **c** Free-energy landscapes of Cys-185-SOH system and Cys-185-SH system, respectively. Two collective variables (CVs), distances of pair pTyr-527-Arg-175 and pair pTyr-527-Arg-155, are used in metadynamics simulations. Color scale: Cys185-SOH white = 10.0 kcal/mol, blue = 0.0 kcal/mol (left), and Cys185-SH white = 10.0 kcal/mol, blue = 0.0 kcal/mol

Cys-SOH by $H_2O_2$ and demonstrate that sulfenylation of both cysteines contribute to $H_2O_2$-dependent enhancement of Src kinase activity based on combined experimental and computational studies. Recent MD simulation studies have indicated significant structural consequences of sulfenylation of specific cysteines in proteins, e.g., EGFR[27] and monoacylglycerol lipase[43], which are structurally distinct from Cys-185 or Cys-277 in Src. Interestingly, the structural alterations characterized in these MD simulations were attributed to the introduction of electrostatic interactions with nearby amino acids upon formation of a sulfenic acid, consistent with our present studies, which similarly indicate that sulfenylation of Cys-185 and Cys-277 induces altered electrostatic interactions through H-bonding within their respective domains.

Combined crystallography and computational modeling have generated a molecular model for Src kinase activation, involving initial conversion of the autoinhibited kinase to a Tyr-416 unphosphorylated active-like state and subsequent phosphorylation of Tyr-416 resulting in stabilization of the active kinase[8,11,47,48]. Our findings indicating Cys-185 and Cys-277 as principal redox-active targets, and their proximity to regulatory tyrosine residues, respectively Tyr-527 and Tyr-416, suggests that their oxidation directly modulates activation. Cys-277 resides on the solvent-accessible G-rich loop near the A-loop, and is solvent accessible. Computational studies using Markov state models by Roux and colleagues highlight A-loop unfolding as the first step on the pathway to the active-like state, occurring on a time scale of ~100 μs[11,49]. These modeling studies were based on simulations of the Src kinase domain in the absence of the SH2/3 domains. Importantly, our conventional and metadynamics simulations of the entire autoinhibited Src protein show that Cys-277-SOH destabilizes the folded A-loop containing Y416 toward

a more favorable unfolded state on a shorter (~1.5–3.5 μs) timescale, which would imply that oxidation of Cys-277 accelerates the conversion to the active-like state, which is potentially stabilized in the absence of Tyr-416 phosphorylation (Fig. 4). Our MD simulations also indicate that Cys-277-SOH induces A-loop unfolding in the absence of other activating structural changes, such as SH2/3 domain dissociation and αC-helix rotation, which is expected to occur at longer time scales[11]. The importance of Cys-185 in redox regulation of Src activity has not been recognized previously, although it was recently implicated in substrate binding through intermolecular disulfide bonding[50]. This cysteine residue is unique in Src, since all other SFKs contain a serine at the corresponding position (Supplementary Table 1), but proximally located cysteines are found within the phosphotyrosine binding region of other SH2 domains[50]. Previous work has shown that Ala and Ser variations of Cys-185 result in tighter binding of phosphorylated peptides, suggesting that Cys-185-SH modifications impact the binding of the autoinhibitory pTyr-527[51]. Our MD simulations indicating that Cys-185-SOH significantly destabilizes pY527 binding to the SH2 domain (Fig. 5) would suggest that sulfenylation of Cys-185 prevents autoinhibition of Src by lowering the binding affinity of pY527 to the SH2 domain. Additionally, it should be noted that Cys-185 within the autoinhibited structure is obscured from solvent, which limits its accessibility by $H_2O_2$, but is expected to be redox-active due to its proximity to the positive Arg-175. Since Cys-185 is solvent exposed in activated Src (PDB ID 1Y57), it is more likely that oxidation of Cys-185 occurs in cooperation with dissociation of pTyr-527 by alternative mechanisms, and thereby enhances or prolongs Src activation.

Our collective experimental and molecular modeling studies have led us to propose a step-wise mechanism for the regulation

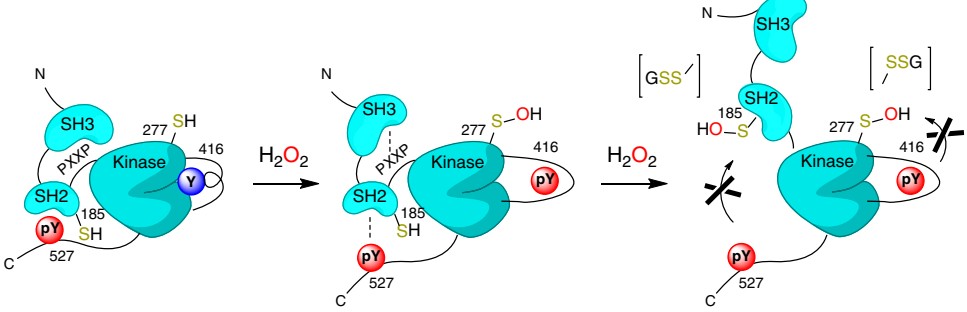

**Fig. 6** Concerted roles of Cys-185 and Cys-277 oxidation induces structural changes that contribute to the activation of Src activity. Initially, the solvent exposed Cys-277 is oxidized to sulfenic acid inducing A-loop unfolding and semi-"unclamping" of the autoinhibited Src structure (middle). The resulting weakened SH2 and c-terminal tail binding enables oxidation of Cys-185 from solvent $H_2O_2$ leading to the full activation of Src (right). It is expected that Cys-185 and Cys-277 sulfenic acids eventually become S-glutathionylated, which permanently obstruct the refolding of Src into the autoinhibited state thereby extending kinase activity

of Src activity by oxidation of Cys-185 and Cys-277 (Fig. 6). Based on our experimental findings indicating Cys-277 as the primary oxidizable cysteine, we propose that initial oxidation of Cys-277-SH triggers the structural conversion of the auto-inhibited state to an active-like state by destabilizing the A-loop helical fold allowing for the exposure of Tyr-416 for phosphorylation. Further structural changes in Src activation, such as rotation of the αC-helix, likely proceed from this state although they were not observed in our MD simulations as they are computed to be beyond our time scale[11]. These structural changes put the kinase into an active-like state in the absence of pTyr-527 dephosphorylation, since the structural changes promoting A-loop unfolding occurred in the absence of dissociation of the autoinhibitory loop and unclamping of the Src protein (Fig. 6 middle). We suggest that unfolding of the A-loop by Cys-277 oxidation may induce semi-"unclamping" of the SH2/3 domains, as the binding of these domains are coupled to the fold of the A-loop[9,47,48], thereby enhancing solvent accessibility of Cys-185 to $H_2O_2$ and facilitating its oxidation. Formation of Cys-185-SOH in turn enhances solvent accessibility of pY527, based on our metadynamics simulations, to promote full dissociation of pY527 from the SH2 domain. Although not addressed in the present study, Cys-SOH species readily reacts with cellular thiols such as glutathione (GSH) to form mixed disulfides (e.g., S-glutathionylated cysteine, Cys-SSG), and our previous studies indicate that $H_2O_2$-mediated activation of Src is associated with sequential formation of Cys-SOH and Cys-SSG[25,42]. Although S-glutathionylation may not directly activate Src[25], we propose that formation of Cys-SSG species on Cys-185 or Cys-277 would induce steric and electrostatic alterations that may enhance structural perturbations. For example, Cys-185-SSG is expected to preclude inactivating interaction of pTyr-527 with the SH2 domain, thereby extending redox-dependent Src activation until Cys-185-SSG is reduced by, e.g., glutaredoxins (Fig. 6 right). Our model therefore suggests that Src activity is controlled by sulfenylation of both Cys-185 and Cys-277 in a cooperative and dynamic mechanism, after which subsequent S-glutathionylation both extends Src activation and facilitates reversal to the inactive state by glutaredoxin-mediated regeneration of reduced cysteines.

It is important to highlight some limitations of our present studies. First, our MD simulations were based on protonated forms of Cys-SOH, as this is the initial product of reaction of Cys-S⁻ with $H_2O_2$ justified by benchmarked $pK_a$ values for simple models due to the variability in experimental estimates.[44,45] However, in light of more recent biochemical estimates of the Cys-SOH $pK_a$ nearer or below physiological pH,[38,46] we cannot rule out that Cys-SOH may also exist in its deprotonated state (or dynamic equilibrium mixture) due to unique and yet uncharacterized second sphere coordination environments of the specific redox-active cysteine residues reported in this work and is the subject of ongoing studies in our labs. We do anticipate that the more anionic Cys-SO⁻ would induce alternative or additional structural changes. Another limitation is that this study did not address the role of other Src Cys residues implicated in previous studies[29,32,52]. Our MS studies indeed indicate that other cysteines (e.g., Cys-238, Cys-498) may also be oxidized to Cys-SOH. Moreover, our relative inability to trap Cys-SOH with either dimedone or BCN on these other cysteines does not necessarily mean that these cysteines are not oxidized, but might instead be a result of their rapid condensation to disulfide species. Indeed, studies with recombinant Src indicated the formation of DTT-reducible higher molecular weight species in the presence of high doses of $H_2O_2$ reflecting formation of inter- or intramolecular disulfides. Chiarugi and co-workers suggested that redox-dependent Src activation during cell adhesion depends on the presence of Cys-245 and Cys-487 and proposed that oxidation of these cysteines may induce structural alterations via an intramolecular disulfide[29], although this seems highly unlikely based on long distances (~30–60 Å) between these cysteine residues in existing structures[9]. Other studies have implicated Cys-498 as a target for S-nitrosylation in activation of Src, although the molecular mechanism was not addressed[52]. Our MD simulations of Cys-498-SOH indicate minimal structural changes in the kinase domain C-lobe, but since Cys-498 is located near the myristate binding pocket within the Src kinase domain,[53] it is plausible that oxidation of Cys-498 might interfere with Src myristoylation. Alternatively, it is possible that these alternative Cys oxidations may regulate Src-dependent signaling in a cellular context by enhancing interactions with regulatory proteins or substrates through, e.g., disulfides, as was recently demonstrated for Cys-185[50], rather than by inducing structural transitions that would promote intrinsic kinase activity. Finally, the field of thiol-based redox signaling has recently recognized that protein cysteines can also exist in the form of persulfides (i.e., Cys-SS$_n$-H).[54] Indeed, we recently reported that cellular Src may also contain Cys-SS$_n$-H species, which are more susceptible to oxidation compared to corresponding cysteines[55]. The functional consequences of such persulfide oxidation for Src activation are unclear and not addressed in this study, since our experimental approaches included DTT, which would have eliminated such (oxidized) polysulfides prior to analysis. Addressing the relevance of such (oxidized) polysulfide species would require additional

technically challenging[56] experimental approaches and alternative MD simulations.

In summary, we demonstrate that the redox-dependent regulation of the Src kinase involves protein structural alterations induced by the oxidation of two cysteine residues, Cys-185 and Cys-277, to intermediate sulfenic acids. Cys-277 is conserved in other SFKs such as Yes and Fgr, which may be subject to similar redox-dependent activation, but Cys-185 is not found in other SFKs (Supplementary Table 1). Therefore, the redox-dependent mechanisms of kinase activation may be unique to Src compared to other SFKs, and may be informative with respect to drug development efforts toward selective Src inhibitors. Current Src inhibitors in clinical development are relatively non-selective since they target the kinase ATP-binding pocket[57]. Efforts to develop covalent Src-selective inhibitors targeting Cys-277 enhance selectivity, but may be compromised upon oxidation of Cys-277[58]. Instead, approaches to target the oxidized states of these cysteines may enhance efficacy and selectivity. Finally, targeting of oxidant-induced structural changes with allosteric inhibitors, as opposed to ATP competitive inhibitors, may offer further selectivity for situations with pathologically elevated ROS[59]. Therefore, a more complete understanding of the molecular details of redox-dependent mechanisms of the SFKs as well as other redox-sensitive kinases[18,19] may offer exciting new directions for drug discovery.

## Methods

**Src expression and purification**. Recombinant Src and variants were expressed in BL21-AI *Escherichia coli* (Thermo) containing the pEX-Src-C-His (Origene, Blue Heron Biotech) and purified in a modified procedure as previously described[60] (see Supplementary methods for complete details).

**Mass spectrometry**. Recombinant Src (~20 μg in dimedone experiments, ~13 μg in BCN experiments, final DTT concentration 0.03–0.25 mM) was incubated with either 1.0 mM 5,5′-dimethyl-1,3-cyclohexanedione (dimedone; Sigma), 100 μM 9-hydroxymethylbicyclo[6.1.0.]nonyne (BCN, Sigma), or 5 mM 4-chloro-7-nitrobenzofuran (NBD-Cl), and then reacted with 0–1.0 mM $H_2O_2$ (Sigma) for 1 h at 25 °C. For quantitative analysis of sulfenylation, Src (~20 μg) was reacted with 0 or 1.0 mM $H_2O_2$ in the presence of light ($d_0$) or heavy ($d_6$) dimedone[40], respectively, for 1 h, followed by catalase quenching. A 1:1 mixture of $d_0$ and $d_6$ labeled samples was run on 10% SDS-PAGE gels and protein bands were excised, washed, and trypinized (Promega V511A). Peptides were reconstituted with 2.5% $CH_3CN$/2.5% formic acid and analyzed by capillary LC–MS/MS on a LTQ (for initial studies with dimedone and NBD-Cl) or a Q-Exactive mass spectrometer coupled to an EASY-nLC (Thermo Fisher Scientific, Waltham, MA, USA). Product-ion spectra were searched using the SEQUEST search engine implemented on the Proteome Discoverer 1.4 (Thermo Fisher). Percolator node was included in the workflow to limit false-positive rates to less than 1% in the data sets. Scaffold 4.05 (Proteome Software) was used for sequence annotation. For experiments involving quantification of dimedone-$d_0$ and dimedone-$d_6$ labeled peptides, targeted mass spectrometry experiments were performed. MS data were acquired with alternating MS–SIM scans and PRM (two scan groups in the Method). Search files (.msf) were then imported into Skyline for selecting the precursor or transitions for quantitation. Four to six transitions were selected from, if possible, the third to the last fragment ions in the product-ion spectrum for quantification. Boundaries of integration were manually evaluated.

**Tyrosine kinase activity measurements**. Tyrosine kinase activity of recombinant Src (1 ng in 15 μL with 2.0 mM DTT) was analyzed using the ADP-Glo (Promega) assay kit according to the manufacture's protocol in a final reaction volume of 25 μL (Supplementary Information). Src was treated with $H_2O_2$ (Thermo) for 15 min prior to initiating catalysis for 60 min at 25 °C within the linear kinetic range. Reactions were measured for luminescence and specific activity values were obtained from an ADP (Promega) linear standard curve.

**H292 cell transfection with Src-FLAG constructs**. Src-FLAG constructs were transiently transfected into NCI-H292 pulmonary mucoepidermoid cells (ATCC) at 85–90% confluence using the Turbofect reagent (Thermo) according to a modified version of the manufacturer's protocol. Briefly, 1 μg of pCMV6-Src (Origene) was mixed with 8 μL Turbofect and 100 μL serum-free RPMI media (per well) and allowed to incubate at RT for 20 min. Cells were changed to 900 μL serum-free media prior to dropwise addition of 100 μL transfection mix. Cells were then incubated at 37 °C for 24 h, followed by PBS washing and 24 h of full RPMI.

Prior to experiments (~48–72 h post-transfection), cells were switched to serum-free RPMI for overnight incubation. H292 cell lines stably expressing C-terminal FLAG-tagged Src were generated by transfecting ~70% confluent cells with 1 μg of DNA from the pCMV6-Src (Origene) and 2 μL of Turbofect Transfection reagent (Fisher) in RPMI media for 24 h as per manufacture's protocol, and selection over 10–15 days in serum containing media with 150 μg/mL of neomycin (G418; Calbiochem).

**Cell culture and treatments**. NCI-H292 cells, a human pulmonary mucoepidermoid carcinoma cell line, were originally obtained from the American Type Culture Collection (ATCC), grown in RPMI 1640 medium containing 10% fetal bovine serum and 1% penicillin/streptomycin at 37 °C and 5% $CO_2$, seeded at 100,000 cells/well in 24-well plates (Corning), and serum starved overnight.[22,25] For in situ analysis of protein sulfenylation (−SOH), cells were preloaded with 5 mM DYn-2 reagent[21] (Kerafast) for 15–30 min, and subsequently stimulated with ATP (Sigma, St. Louis, MO; 100 μM) for 10 min. Cells were then lysed in chelator-free HEPES lysis buffer and clicked to biotin-azide (Kerafast) using established protocols (Supplementary Information). As an alternative approach to detect sulfenylated proteins, cells were lysed with western solubilization buffer containing 1 mM DCP-bio1 (Kerafast), as described previously[24,25]. FLAG-tagged Src and mutants were purified using Anti-FLAG M2 magnetic beads (Sigma) and eluted with 3× FLAG peptide (Sigma) as per manufacture's protocol (Supplementary Information).

**Western blotting**. Derivatized cell lysates were separated by 10% SDS-PAGE gels and transferred to nitrocellulose membranes, and probed using antibodies against p-Src Tyr 416 (1:400; 2101S), p-Src Tyr 527 (1:400; 2105S), Src (L4A1; 1:1000; 2110S), FLAG (DDK; 1:1000; 2368S) (Cell Signaling), or streptavidin peroxidase polymer ultrasensitive (1:10,000; S2438; Sigma). Primary antibodies were probed with rabbit- or mouse-specific secondary antibodies conjugated with HRP (Cell Signaling) and detected by enhanced chemiluminescence (Pierce). Membranes were imaged with Amersham Imager 600 (GE Healthcare) and band densities were quantified using ImageQuant TL (v8.1.0.0).

**MD simulations**. All molecular dynamics and metadynamics simulations were performed with the CHARMM36 force field[61] in addition to our customized parameters for the cysteine sulfenic acid (Cys-SOH). The parameters (i.e., for bonds, angles, and dihedral angles) and partial charges for cysteine sulfenic acid were validated or modified according to the density-functional theory calculations with B3LYP/6-31g** in Jaguar[62,63] (see Cysteine sulfenic acid force field parameters of Supplementary methods). The protein models were constructed from the autoinhibited Src structure (PDB ID: 2SRC)[9]. A web-based graphical user interface CHARMM-GUI[64] was used to prepare the initial models for simulations, which contain our protein models, 17,000–26,000 TIP3P water molecules, chloride potassium as counter ions, totaling near 58,400–85,400 atoms in a periodic box 85 × 89 × 82–94 × 94 × 94 Å³. After the equilibrium stage[65] for 10 ns in the NAMD[66] program, production runs of conventional MD simulations (for 2–5 μs) were carried out on the specialized ANTON supercomputer using the software program Anton 2.13.0[67] and with the Desmond program (Schrödinger, Inc.) on graphical processing units (GPUs). Metadynamics simulations were also carried out on GPUs to determine the two-dimensional, free-energy landscapes of conformational transition in the cysteine oxidation region in comparison with corresponding nonoxidized cysteines, using two CVs that were chosen based on conformational changes suggested by the conventional MD simulations. To ensure convergence, we have calculated the free-energy difference every 10 ns after 80 ns: the key energy basins remain to the final 120 ns with a general SD of 0–0.68 kcal/mol for the maximum free-energy difference. A summary of our simulations is provided in Supplementary Table 3.

**Statistical analysis**. Data are expressed at means ± s.e.m. for the indicated number of observations and comparisons are analyzed with student's *t* test or two-way ANOVA.

## Data availability

The data that support the findings of this study are available from the corresponding authors upon request.

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

## Acknowledgments

Research support was provided by National Institutes of Health grants R01 HL085646, R01 HL138708, and R01 ES021476 (to A.v.d.V.) and postdoctoral fellowships F32 HL129706 (to D.E.H.) and T32 HL076122 (to C.M.D. and A.C.L.). Anton computer time was provided (to A.v.d.V.) by the Pittsburgh Supercomputing Center (PSC) through NIH Grant R01 GM116961. The Anton machine at PSC was generously made available by D.E. Shaw Research. Computational resources were provided (to J.L.) by Vermont Advanced Computing Core (VACC) and Stampede from the Extreme Science and Engineering Discovery Environment (XSEDE, NSF Grant No. ACI-1053575). The Proteomics Facility is supported by the Vermont Genetics Network through NIH Grant 8P20 GM103449. National Center for Research Resources (NCRR 5 P30 RR 032135) and the National Institute of General Medical Sciences (GMS 8 P30 GM103498) from the NIH, University of Vermont Neuroscience COBRE Grant (to R. Parsons). Confocal microscopy was performed on a Zeiss 510 META laser scanning confocal microscope supported by NIH grant 1S10 RR019246 from the National Center for Research Resources. We thank Edward Zelazny and Thomas Buttolph III for technical expertize in DNA preparation.

## Author contributions

A.v.d.V., J.L. and D.E.H. conceived and designed experiments and modeling approaches; D.E.H., C.M.D., M.H., C.V., A.C.L., B.A.A. and S.L.W. performed the experiments; C.L. and D.E.H. performed molecular dynamics and metadynamics simulations; D.E.H., C.M.D., C.L., B.D., Y-W.L., J.L. and A.v.d.V. analyzed the data; D.E.H., C.M.D., C.L., Y-W.L., J.L. and A.v.d.V. co-wrote the paper.

## Additional information

**Competing interests:** The authors declare no competing interests.

