## [Peer Review File · Nature Communications]

Reviewers' comments:

Reviewer #1 (Remarks to the Author):

The study by van der Vliet and colleagues tells an important story regarding redox regulation of Src kinase. Prior work in this area has been murky and this piece is one of the first to provide aspects of molecular clarity. The impact, as the work currently stands, is moderate or at the level of JBC. In order to elevate this excellent and careful study to the level of Nature Communications the following should be considered: 1) Although Biotin-DCP is a solid reagent, it cannot probe in situ oxidation state (the biotin moiety is well known to block permeability) and the biotin fragmentation within the MS/MS spectra precludes robust, confident LC-MS/MS assignments. The mass spectra, while plausible in some cases, are less than ideal. In addition, this journal has published the current state-of-the-art with respect to quantitative analysis of S-sulfenylation. Therefore, to accept less technology would be going backwards, instead of driving the field forward. 2) To address other redox-related sites in Src kinase there are also faster, commercially available, reagents that would address competition and serve to further clarify redox-regulation of Src kinase. 3) The pKa of sulfenic acids has been reported in proteins and peptides as 7 or less. Therefore, it would be important to complement current dynamic analyses with thiolate and sulfenate models. In sum, I am highly supportive of this exciting and important work. A few straightforward experiments with modern technology and, comprehensive analysis of protonation states in MD, would suffice. This review was prepared by Kate Carroll of Scripps Florida and is not intended to be blind.

Reviewer #2 (Remarks to the Author):

Authors have investigated the redox regulation of Src kinase via oxidation of cysteines. In particular, they report Cys-185 and Cys-277 play a role in regulation of Src activity. MD simulations indicate that oxidation of Cys-277 promotes rapid unfolding of activation loop and Cys-185 increases the solvent accessibility near Tyr-527, which could help in release of regulatory SH2 and SH3 domains from the catalytic domain.

There are several key issues, which should be addressed by the authors:

- 1) Authors mention that Cys 498 could have an allosteric effect on the Src activity. However, they do not provide any reason or basis for this observation. Cys 498 is near the Myristol binding pocket, which is known to influence the activity of Src and Abl kinases. Authors could analyze their existing simulation and a simulation with Cys498-OH to show that the cross-coupling between the residues near Cys498 and the activation loop or other functional regions of the catalytic domain is altered due to the oxidation. I do not recommend the authors to do this experiment but the allosteric effects could be captured using HDX-MS experiment as outlined in Lacob et al., PLoS one, 6(1): e15929. 2011.
- 2) I understand that authors have used force field parameters from CGenFF. I appreciate that they have shared these parameters in SI. However, it would have been helpful to perform some elementary parameter validation before using and sharing these parameters with the scientific community.
- 3) Authors claim that Cys277 oxidation leads to rapid unfolding of the activation loop. I am concerned that this claim is not supported by their simulation data. I find it surprising that oxidation of a solvent-exposed cysteine leads to complete unfolding of A-loop. The unfolding timescale for A-loop (from Meng et al., PNAS, 113, 33, 2016 and Shukla et al., Nat. Commun., 5: 3397, 2014) is shown to be in the range 1-10 microseconds. Authors have performed simulations of length 1.6 s without CYS-OH

and 5.5 s with Cys277-OH. It is not entirely surprising that they observe A-loop unfolding in the 5.5 s simulation. A single trajectory of few s is not sufficient for obtaining statistics that can validate the claim made in the manuscript.

4) The mechanistic explanation for the effect of Cys277 and Cys185 oxidation are not grounded in rigorous quantitative analysis and simulations. Authors could have focused on estimating the change in the free energy barrier due to the oxidation by estimating the change in the free energy of unfolding of the A-loop or unclamping of SH2 domain as outlined in reference 9 and 10.

5) There are several minor typos and corrections in the manuscript, an incomplete list is included below.

- a. Page 1, Abstract: "in response to NADPH" instead of "in response NADPH"
- b. Page 2, Introduction: provide the organism name instead of "avian sequence number"
- c. Page 2, Introduction: "conformation" instead of confirmation.
- d. Page 3, "presence of dimedone" instead of "presence dimedone"
- e. Page 13, Figure 2D is missing or Figure 2 is incorrectly labeled. I also don't understand the reason for the two dark bands in Figure 2A (sub figure in row 2, column 3, Ctrl)

We much appreciate the encouraging and constructive comments by the reviewers, and the opportunity to address these comments and resubmit our manuscript. We have performed substantial additional experimentation, which include additional MS analyses, cell labeling studies, as well as MD simulations and metadynamics analyses, in order to address key comments by both reviewers. We are pleased that these additional studies confirmed and substantiated our original conclusions, and believe that inclusion of these additional data and further strengthen our manuscript. As a result, extensive revisions have been made to manuscript, which are all highlighted in red font. Because of the substantial additional experimental and computational input by Christopher Dustin and Chenyi Liao (2nd and 3rd author, respectively) during this revision stage, we chose to recognize their additional efforts by state that their contributions were equivalent. Also, because of the significant contributions of Dr. Heppner in initiating certain aspects of this work, and working closely with Mr. Dustin and Dr. Liao during these revisions, we chose to list him as a co-corresponding author. We kindly thank the reviewers for their insightful comments, and present a point-by-point response to the reviewers' comments below.

Reviewer 1:

1) Although Biotin-DCP is a solid reagent, it cannot probe in situ oxidation state (the biotin moiety is well known to block permeability) and the biotin fragmentation within the MS/MS spectra precludes robust, confident LC-MS/MS assignments. The mass spectra, while plausible in some cases, are less than ideal. In addition, this journal has published the current state-of-the-art with respect to quantitative analysis of S-sulfenylation. Therefore, to accept less technology would be going backwards, instead of driving the field forward.

Response: We agree with the reviewer on these points. We'd like to point out that our MS analyses were based on using dimedone rather than DCP-Bio1, so this should not be a concern. With respect to our use of DCP-Bio1 to probe cysteine oxidation in situ (i.e. in intact cells), we concur that cellular permeability may be limited and we therefore used this probe only at the time of cell lysis, under conditions that minimize artefactual oxidation and/or thiol exchange, as we and others have performed previously. While this approach may also capture some artefactual sulfenylation during cell lysis, it clearly demonstrated marked differences resulting from cell activation, as we had demonstrated in previous studies ^{1,2}. Nevertheless, to address the reviewer's concern, we also used the cell-permeable probe (DYN-2) to assess cysteine sulfenylation in situ (i.e. by preloading cells prior to cell stimulation) using click chemistry to biotin-tagged modified cysteines for avidin purification. Indeed, this yielded highly similar results, and confirmed that ATP-induced sulfenylation and phosphorylation of Flag-tagged Src protein constructs expressed in H292 cells was markedly attenuated in C185A or C277A mutants compared to wild-type protein. We include these new findings in a new Fig. 3, and in Supplemental Fig. S9, and also relegated our original findings using DCP-Bio1 to the Supplemental Information in Fig. S10. While performing these studies, we recognized some differences in expression levels of the various Src constructs in stably transfected cell lines, and therefore also performed similar studies with transient transfections of these Src constructs, with similar results. We would also like to point out that the in situ labeling approach with DYN-2 has the potential to interfere with redox signaling events (for example by possibly interfering with peroxiredoxin function during cell stimulation), and therefore view both labeling approaches as complementary, and thus chose to include both data sets in the revised manuscript.

The reviewer also commented on the quality of our MS analysis and suggested the use of more state-of-the-art quantitative analysis of Src sulfenylation, based on a method described recently in *Nature Communications*³. We attempted to use this reported procedure to quantitatively evaluate Src cysteine oxidation in a cellular context, but were unfortunately unsuccessful. We suspect that this may be due to the relatively low abundance of Src in our cell system, which would require higher protein input and would become cost prohibitive. Also, it was our goal to address Src oxidation during physiologically relevant cell stimulation by ATP (an important damage signal that activates DUOX1 in airway epithelial cells), which is subtler compared to the direct cell exposure to H₂O₂ used in this former study. To address the relative susceptibility of the various Src cysteine residues, we decided to use a quantitative MS approach, by labeling reduced or oxidized Src with either light or heavy isotopes of dimedone (dimedone-d6), using alternating MS-selected-ion monitoring (SIM) scans and parallel reaction monitoring (PRM) to determine heavy/light ratios of modified peptides. These findings are now reported in a new Figure 1 and Figure S5, which show MS1 spectra, extracted ion chromatograms, and MS/MS spectra for the two major cysteines modified, and also included a new Table 1 that summarizes observed heavy/light ratios for other modified peptides. Full data for all replicate analyses are included in Supplemental Dataset 4. Based on the relative increases in H/L ratios, and the reproducibility between replicate analyses (low standard deviations), we concluded that Cys-277 and Cys-185 are the most prominently modified residues, which further justifies our focus on these residues in our subsequent studies. However, we do recognize that other cysteines may also be subject to oxidation and may affect Src activation, and have attempted to address this with additional MD simulations and discussion.

2) To address other redox-related sites in Src kinase there are also faster, commercially available, reagents that would address competition and serve to further clarify redox-regulation of Src kinase.

Response: We recognize that reaction of dimedone with sulfenic acids is rather slow, and chose to extend analysis of Src cysteine oxidation by using bicycle[6.1.0]nonyne (BCN) as an alternative sulfenyl probe, which is known to react approximately 100 times faster than dimedone. This again confirmed C185 and C277 as the major sites of sulfenylation by H₂O₂, although it also revealed modification of an additional cysteine (C245), although no labeling of other cysteines was detected. We have included these findings in Supplemental Table S2 and Supplemental Dataset 2, and rationalized that these findings with BCN labeling further support our focus on C185 and C277 as the main sites for sulfenylation by H₂O₂.

3) The pK_a of sulfenic acids has been reported in proteins and peptides as 7 or less. Therefore, it would be important to complement current dynamic analyses with thiolate and sulfenate models.

Response: This is an excellent point. Even though the original reaction product of a thiolate with H₂O₂ is a sulfenic acid, potential deprotonation may occur to affect outcomes. We chose to address this issue by using pK_a calculations in Jaguar (Schrödinger Inc.) with a DFT method (the B3LYP functional and the 6-31G** basis set) to determine pK_a values of ethyl-thiol and the corresponding sulfenic acid, which yielded values of 10.3 and 8.7, respectively. We cannot say to what extent this applies to cysteine thiols or sulfenic acids within Src (which could in fact be

lower in certain cases), but feel that this justifies performing our MD simulations based on the protonated forms. We modified the manuscript on page 6 to include these calculations and discussion. We also modified the Discussion to clarify the potential limitation of not having modeling the deprotonated forms.

Reviewer 2:

1) Authors mention that Cys 498 could have an allosteric effect on the Src activity. However, they do not provide any reason or basis for this observation. Cys 498 is near the Myristol binding pocket, which is known to influence the activity of Src and Abl kinases. Authors could analyze their existing simulation and a simulation with Cys498-OH to show that the cross-coupling between the residues near Cys498 and the activation loop or other functional regions of the catalytic domain is altered due to the oxidation. I do not recommend the authors to do this experiment but the allosteric effects could be captured using HDX-MS experiment as outlined in Lacob et al., PLoS one, 6(1): e15929. 2011.

Response: We thank the reviewer for these valuable suggestions. We identified Cys-498 as an oxidizable cysteine in our original dimedone trapping experiments, but (somewhat surprisingly) did not detect Cys-498 trapping with BCN. Therefore, we have de-emphasized Cys-498 as a potentially important oxidant-sensitive cysteine. We have, however, performed additional MD simulations of Cys498-SOH using the same approach for the other cysteine residues, and observed only minimal conformational changes at the helices bundles where Cys-498 is located. This is now included in Supplemental Fig. 14A and B. In addition, metadynamics simulations show comparable energy basins for the Cys498-SH and Cys498-SOH systems, suggesting that the oxidation of Cys-498 has a minor impact on the dissociation of the C-lobe, which is reported in Supplemental Fig. 14C. Nevertheless, we agree with the reviewer that oxidation of Cys-498 could potentially interfere with Src regulation by myristoylation, since a myristate binding pocket is present within this region of the kinase domain⁴ and added this suggestion to the discussion (on page 10).

2) I understand that authors have used force field parameters from CGenFF. I appreciate that they have shared these parameters in SI. However, it would have been helpful to perform some elementary parameter validation before using and sharing these parameters with the scientific community.

Response: We agree with the reviewer that parameter validation is essential in this work. Therefore, we have carefully examined the structural parameters (*i.e.* for distances, angles, and dihedral angles) and partial charges for the –S-O-H moiety, which were initially obtained from the CHARMM General Force Field (CGenFF) program, according to our DFT optimized models. In addition, we also carried out elementary validation, *i.e.*, dihedral validation, using quantum calculations at the DFT level (Figure S11).

The bond-stretching and angle-bending parameters obtained from CGenFF are reliable with penalties below 10, while the parameters for dihedral angle –C-S-O-H require further validation. Therefore, we performed a relaxed scan for dihedral angles at a 10-degree increment using the program Jaguar (Schrödinger, *Inc.*). Each conformer is optimized with energy calculations at the quantum and force field levels.

We used the CH₃CH₂-SOH model system to compare our parameters for -SOH with DFT calculations geometrically and energetically in relaxed dihedral scanning. We scanned the dihedral angle –C-S-O-H from 0 degrees to 180 degrees, over 10-degree increments from point to point in implicit water using the Poisson-Boltzmann model. At each scan point, a series of single-point energy calculations using DFT with B3LYP/6-31g** was applied while geometry optimization was also carried out, which is a general force-filed parameterization scheme.⁵ The

total energy difference by relaxed dihedral scanning using DFT calculations was displayed in blue line in Figure S11.

The dihedral term in the CHARMM force field was given by,

$$V(\phi) = K_{\phi}[1 + \cos(n\phi - \phi_0)] \quad (s1)$$

K_{ϕ} is the dihedral force constant, n is the multiplicity of the function, ϕ_0 is an equilibrium angle. The best fitting dihedral parameter is: K_{ϕ} , n , and ϕ_0 values as 3.2, 2, and 0, respectively. Here, we chose K_{ϕ} of 1.1, referring to other value given to similar groups used in the CHARMM force field, for example, -C-C-O-H; the lower energy barrier may contribute to faster dynamics and easier rotation for -C-S-O-H, within conformational error range. The dihedral term energy calculated from equation (s1) is shown in orange line in Figure S11, which generated standard deviation of 2.2 kcal/mol. Accordingly, the total energy difference of each structure generated from Jaguar's relax-scanning were calculated in NAMD program and displayed in red line in Figure S11. In short, our dihedral-angle-dependent energy profile is consistent with DFT calculations, which validate the dihedral angle geometry preference for the sulfenic acid group (-SOH).

3) Authors claim that Cys277 oxidation leads to rapid unfolding of the activation loop. I am concerned that this claim is not supported by their simulation data. I find it surprising that oxidation of a solvent-exposed cysteine leads to complete unfolding of A-loop. The unfolding timescale for A-loop (from Meng et al., PNAS, 113, 33, 2016 and Shukla et al., Nat. Commun., 5: 3397, 2014) is shown to be in the range 1-10 microseconds. Authors have performed simulations of length 1.6 microsecond without CYS-OH and 5.5 microsecond with Cys277-OH. It is not entirely surprising that they observe A-loop unfolding in the 5.5-microsecond simulation. A single trajectory of few s is not sufficient for obtaining statistics that can validate the claim made in the manuscript.

Response: We appreciate these comments about the Src A-loop. We examined recent related publications more carefully and compared to our conventional MD (2 replicas of 10 microseconds in total) and metadynamics simulations of the Cys277-SH and Cys277-SOH systems. Both conventional simulations indicate clear loss in helicity within 5 microseconds (which is now shown in revised Fig. 4B). In addition, our metadynamics simulations show that unfolded conformations of the A-loop (yellow in Figure 4A) are energetically favorable after Cys277-SH is substituted for Cys277-SOH. These findings are now included in the revised manuscript in Fig. 4C.

4) The mechanistic explanation for the effect of Cys277 and Cys185 oxidation are not grounded in rigorous quantitative analysis and simulations. Authors could have focused on estimating the change in the free energy barrier due to the oxidation by estimating the change in the free energy of unfolding of the A-loop or unclamping of SH2 domain as outlined in reference 9 and 10.

Response: We agree with the reviewer that quantitative evidence is critical to support our conclusions. Thus, we have carried out metadynamics simulations for Cys185-SOH, Cys277-SOH in comparison with their corresponding cysteine controls. The collective variables (CVs) were chosen based on conformational changes observed from the conventional simulations. For

example, to evaluate sulfenylation of Cys277, we chose the pair distance of Y416 - D386 and pair distance of R419 - D386 as two collective variables as large increase of pair distance in either of the variable will ultimately lead to unfolding of the A-loop. For sulfenylation of Cys185, we chose pair distance of pTyr-527 - R175 and pair distance of pTyr-527 - R155 as two collective variables because the increase in pair distance of both will lead to the dissociation of pTyr-527 of the C-terminus. As shown, modification of Cys185, Cys277 to their corresponding sulfenic acids did in both cases alter the free-energy landscapes, which contribute to the unfolding of the A-loop or dissociation of the C-terminus, respectively. These results are now included in newly added Figures 4 and 5, and clarified in the main text.

5) There are several minor typos and corrections in the manuscript, an incomplete list is included below.

Response: We are thankful to the reviewer for pointing out these typos and corrections, and have made the appropriate changes.

References:

1. Heppner DE, Hristova M, Dustin CM, Danyal K, Habibovic A, van der Vliet A. The NADPH oxidases DUOX1 and NOX2 Play Distinct Roles in Redox Regulation of Epidermal Growth Factor Receptor Signaling. *Journal of Biological Chemistry* **291**, 23282-23293 (2016).
2. Habibovic A, *et al.* DUOX1 mediates persistent epithelial EGFR activation, mucous cell metaplasia, and airway remodeling during allergic asthma. *JCI Insight* **1**, e88811 (2016).
3. Yang J, Gupta V, Carroll KS, Liebler DC. Site-specific mapping and quantification of protein S-sulphenylation in cells. *Nature communications* **5**, 4776 (2014).
4. Patwardhan P, Resh MD. Myristoylation and membrane binding regulate c-Src stability and kinase activity. *Molecular and cellular biology* **30**, 4094-4107 (2010).
5. Wildman J, Repiščák P, Paterson MJ, Galbraith I. General Force-Field Parametrization Scheme for Molecular Dynamics Simulations of Conjugated Materials in Solution. *Journal of Chemical Theory and Computation* **12**, 3813-3824 (2016).

Reviewers' comments:

Reviewer #1 (Remarks to the Author):

The authors have made commendable improvements to the manuscript and addressed many reviewer concerns. It is an important story. However, several issues still require attention, as follows:

1. Regardless of what probe was used for the MS/MS studies, the key issue here is data quality. First, a visual inspection of primary MS/MS spectra supplied in the Supplementary Figures indicates variation in quality and therefore, certainty. Second, a rigorous discussion of relevant statistical parameters and how MS/MS data were vetted (beyond search parameters) is lacking. Various Sciquest parameters are included in Datasets, but values used to quality control should be defined and the "ions value" matching experimental ions to predicted ions doesn't appear to be presented. Finally, the presentation of the MS/MS data in the Supplementary Datasets might be improved by ordering peptides with respect to confidence and potentially highlighting cysteines in bold.

2. DYn-2 [Paulsen et al. *Nat Chem Biol* 8, 57-64 (2012)] and d6-dimedone [Seo and Carroll *Angew Chem Int Ed* 50, 1342-5 (2011)] need to be appropriately referenced the first time they are mentioned in the manuscript.

As an aside, our *Nat Comm* paper also reported an analysis of dynamic changes in EGF-mediated S-sulfenylation, not just H₂O₂ treatment. The correlation between the two stimulants was excellent, though the EGF experiments demonstrated more selectivity, as expected. Regarding the potential concern that in situ by DYn-2 could inhibit signaling pathways, reaction kinetics suggest that this particular probe samples <5-10% of cellular sulfenic acids, which makes it an unlikely or weak inhibitor at best (Gupta, V and Carroll, *KS Biochim Biophys Acta* 1840, 847-875 (2014)). Regardless, if this is truly a serious concern for the authors, they could examine this possibility by monitoring Src-phosphorylation (or tyrosine phosphorylation more broadly) as a function of probe concentration.

3. BCN and other strained cyclic alkynes are known to cross-react with thiols (Galardon and Padovani *Bioconjug. Chem.* 26, 1013-1016 (2015) also Gupta and Carroll *Chem Sci* 7, 400-415 (2016)). In this light, BCN was probably not the best choice and faster alternatives with demonstrated selectivity are available (Gupta et al. *J. Am. Chem. Soc.* 139, 5588-5595 (2017)). Regardless, my understanding is that BCN did not label Src in the absence of H₂O₂ (or at least high confidence peptides were not identified). The authors should include a brief statement regarding the established cross-reactivity of BCN (cautioning use in gel-based studies), but which can be addressed in MS/MS ensuring the sulfoxide (+16 Da) relative to the thiol adduct.

4. In light our published data, indicating the pK_a of sulfenic acid in a dipeptide ~7 [(Gupta and Carroll *Chem Sci* 7, 400-415 (2016))] and Poole's excellent study that experimentally measures the pK_a in four proteins [(pK_a 5.9-7.2; Portillo-Ledesma et al. *Biochemistry* 57, 3416-3424 (2018))] resorting to calculated values for ethyl-thiol and related sulfenic acid appears a bit silly. These states are most certainly in equilibrium at physiological pH and understanding how trajectories change in the sulfenate are relevant. Regardless, the need for additional calculations can be mitigated by clearly citing experimentally measured values, which differ from model calculations (they might consider taking these calculations out altogether), and the statement pertaining to limitations of modeling only RSOH statement kept in place.

5. The discussion section should be evaluated carefully to avoid simple restatement of results. In particular, a short compare/contrast of findings from prior MD simulations of different EGFR redox states is lacking (regardless of different cysteine locations). It is important that the reader know that

there is a history of using MD to assess conformational change with cysteine oxidation, in my lab, and others. Also, experimental support for the cooperative or sequential oxidation mechanism in Fig. 6 could be further clarified.

6. Finally, in light of the authors' recent manuscript raising the specter that signal from dimedone or related probes may, in fact, represent oxidized persulfides (RSSOH) masquerading as RSOH (at least in gels, since MS/MS clearly distinguishes these states) this point must be squarely addressed. The ways in which this possibility was experimentally ruled out should be briefly mentioned in the main text.

This review was prepared by Kate Carroll at Scripps Florida and not intended to be blind.

Reviewer #2 (Remarks to the Author):

Authors have addressed all of my comments adequately in their response. I recommend publication of this manuscript.

We much appreciate the positive and encouraging comments by both reviewers regarding our revised manuscript, and the suggestions by reviewer 1 to further clarify some key aspects and some additional points of discussion. We have reanalyzed some of our MS data, and have made several revisions and additions in the text (highlighted in red font) to address the various comments, which also included some additional references. We also updated our Supplemental Information and Supplemental Datasets, based on reviewer's suggestions. Per journal policy, we have also converted quantitative graphed data into box plots. With respect to Fig. 2, we have chosen to perform additional correlation analyses (which is included in a new Figure S7) to strengthen our overall conclusion with respect to H₂O₂ dependent increase in Src activity. We also noted a minor error in our original quantifications of the blots in Fig. 3, and have reanalyzed them to generate revised Fig. 3B. We wish to kindly thank the reviewer for the additional insightful comments, and present a point-by-point response below:

1. Regardless of what probe was used for the MS/MS studies, the key issue here is data quality. First, a visual inspection of primary MS/MS spectra supplied in the Supplementary Figures indicates variation in quality and therefore, certainty. Second, a rigorous discussion of relevant statistical parameters and how MS/MS data were vetted (beyond search parameters) is lacking. Various Sciquest parameters are included in Datasets, but values used to quality control should be defined and the "ions value" matching experimental ions to predicted ions doesn't appear to be presented. Finally, the presentation of the MS/MS data in the Supplementary Datasets might be improved by ordering peptides with respect to confidence and potentially highlighting cysteines in bold.

Response: We apologize for this confusion, which stems from the fact that different data sets were acquired on different instruments. Initial exploration MS studies with dimedone and NBD-Cl were performed using a linear ion trap (LTQ) instrument, whereas the more recently acquired data using BCN and the quantitative analysis using dimedone-d6 were performed using a more high-resolution Q-Exactive mass spectrometer. This contributed to the apparent variation in quality of MS/MS spectra presented in Figure S2 and S4. We also noted an error in Table 1 with respect to the reported ΔM values, which should have been listed as ppm in case of DCN and NBD-Cl. We have clarified this issue more clearly in the Methods section, and in the Supplementary Information, and also corrected Table 1. Importantly, peptides were all identified within a dataset limited to 1% FP when searched against a target-decoy database using percolator in the search workflow.

To further help clarify the reviewer's concern, we also decided to include MS/MS spectra from the targeted analysis of dimedone trapping in a new Supplementary Dataset 5, showing high quality spectra for all the dimedone-d6 labeled peptides. Please note that all these experiments were performed on an in vitro reaction with only one single protein Src in the solution. The fact that all the measured precursor masses of the identified Src peptides were within 0.5 ppm of the theoretical masses and that the tandem mass spectra (MS/MS) exhibit a continuous stretch of y-ion series, and clear peak assignments, indicates confident identifications.

With respect to the last point in this overall comment, we have now highlighted all detected modifications on cysteines and methionines more clearly, by highlighting the involved residues in bold red font in the Supplemental Datasets. Although all listed peptides were denoted as high quality based on Proteome Discover (we have eliminated medium/low confidence peptides that were included originally in some tables), we have also followed the reviewer's suggestion by ordering the peptides based on descending Xcorr values (as a measure of overall confidence).

2. DYn-2 [Paulsen et al. Nat Chem Biol 8, 57-64 (2012)] and d6-dimedone [Seo and Carroll Angew Chem Int Ed 50, 1342-5 (2011)] need to be appropriately referenced the first time they are mentioned in the manuscript.

As an aside, our Nat Comm paper also reported an analysis of dynamic changes in EGF-mediated S-sulfenylation, not just H₂O₂ treatment. The correlation between the two stimulants was excellent, though the EGF experiments demonstrated more selectivity, as expected. Regarding the potential concern that in situ by DYn-2 could inhibit signaling pathways, reaction kinetics suggest that this particular probe samples <5-10% of cellular sulfenic acids, which makes it an unlikely or weak inhibitor at best (Gupta, V and Carroll, KS Biochim Biophys Acta 1840, 847-875 (2014)). Regardless, if this is truly a serious concern for the authors, they could examine this possibility by monitoring Src-phosphorylation (or tyrosine phosphorylation more broadly) as a function of probe concentration.

Response: We have included the two references as requested, both in the main text and in the Methods section. With respect to the potential inhibitory capacity of situ DYn-2 labeling, we agree with the reviewer that this is likely not a major concern based on kinetic considerations. However, we brought up this potential concern in our previous response primarily to justify our alternative use of DCP-Bio1 labeling (which is performed at the time of cell lysis, and thus avoids such potential artifact) as a complementary approach. Because both strategies yielded qualitatively similar outcomes, we felt it worthwhile to report results using both complementary strategies, which strengthen our overall conclusion.

3. BCN and other strained cyclic alkynes are known to cross-react with thiols (Galardon and Padovani Bioconjug. Chem. 26, 1013-1016 (2015) also Gupta and Carroll Chem Sci 7, 400-415 (2016)). In this light, BCN was probably not the best choice and faster alternatives with demonstrated selectivity are available (Gupta et al. J. Am. Chem. Soc. 139, 5588-5595 (2017)). Regardless, my understanding is that BCN did not label Src in the absence of H₂O₂ (or at least high confidence peptides were not identified). The authors should include a brief statement regarding the established cross-reactivity of BCN (cautioning use in gel-based studies), but which can be addressed in MS/MS ensuring the sulfoxide (+16 Da) relative to the thiol adduct.

Response: We agree with the reviewer that BCN has the potential downside of being capable of reacting with some cysteines or persulfides. We initially only searched for formation of –S(O)-BCN adducts (ie. a molecular mass increase of 166), which indeed were not found in untreated control samples (except in some cases with low/medium confidence), we now have reanalyzed our data to address potential reaction of BCN with non-oxidized Src protein. However, no high confidence peptides with –S-BCN adducts were detected in any case, and we have now addressed this in the text. We also noted that our MS datasets were not completely clear on this point, and have now modified Table S2 and in Supplementary Dataset 2 to indicate detection of –S(O)-BCN adducts, rather than –S-BCN adducts. We also added a comment in the results section to address this potential caveat, by citing the suggested references.

4. In light our published data, indicating the pK_a of sulfenic acid in a dipeptide ~7 [(Gupta and Carroll Chem Sci 7, 400-415 (2016))] and Poole's excellent study that experimentally measures the pK_a in four proteins [(pK_a 5.9-7.2; Portillo-Ledesma et al. Biochemistry 57, 3416-3424 (2018))] resorting to calculated values for ethyl-thiol and related sulfenic acid appears a bit silly. These states are most certainly in equilibrium at physiological pH and understanding how trajectories change in the sulfenate are relevant. Regardless, the need for additional calculations can be mitigated by clearly citing experimentally measured values, which differ from

model calculations (they might consider taking these calculations out altogether), and the statement pertaining to limitations of modeling only RSOH statement kept in place.

Response: We concur that our calculations of pKa values in ethyl thiol and the corresponding sulfenic acid may not be very meaningful in light of some experimental determinations in some proteins, even though they are in general agreement and justify our choice for modeling the protonated form of Cys-SOH, even though we recognize the potential contribution of deprotonated forms. We have included the suggested citations as recommended by the reviewer, and moved our DFT pKa calculations to the supplementary information. We recognize that the relative amounts of the protonation state of residue side chains are important in the determination of their structural and functional consequences, but in the absence of the precise pKa values of Cys-SH and Cys-SOH in Src we cannot easily determine which species is most relevant. Based on the various experimental estimates and calculated values, as well as previous MD simulations which were also based on the protonated forms (e.g. Scalvini et al., 2016) or indicated structural changes only upon modeling the protonated form (Truong et al., 2016), we felt our choice of modeling protonated Cys-SOH was justified, although we recognize this limitation in the Discussion section.

5. The discussion section should be evaluated carefully to avoid simple restatement of results. In particular, a short compare/contrast of findings from prior MD simulations of different EGFR redox states is lacking (regardless of different cysteine locations). It is important that the reader know that there is a history of using MD to assess conformational change with cysteine oxidation, in my lab, and others. Also, experimental support for the cooperative or sequential oxidation mechanism in Fig. 6 could be further clarified.

Response: We have carefully revised the discussion section to minimize restatements of results. We acknowledge that previous MD studies have addressed the molecular impact of cysteine oxidation, including sulfenylation, and have followed the reviewer's suggestion by discussing other examples of MD simulations of cysteine sulfenylation. Even though they may involve structurally unrelated cysteines, these other studies buttress the point that a small modification such as cysteine sulfenylation has the potential to induce structural alterations, which appear to be related to new electrostatic interactions with nearby amino acids through hydrogen bonding. In addition, the reviewer requested some support for the proposed sequential oxidation mechanism illustrated in Figure 6, which are largely based on our experimental observations that C277 appears to be a primary target for sulfenylation, based on quantitative MS analysis, and can proceed to S-glutathionylation based on kinetic analyses and blocking experiments with dimedone reported previously (Heppner et al., JBC 2016). We have revised the text in the Discussion related to this point, to hopefully clarify and support our proposed mechanism.

6. Finally, in light of the authors' recent manuscript raising the specter that signal from dimedone or related probes may, in fact, represent oxidized persulfides (RSSOH) masquerading as RSOH (at least in gels, since MS/MS clearly distinguishes these states) this point must be squarely addressed. The ways in which this possibility was experimentally ruled out should be briefly mentioned in the main text.

Response: We much appreciate the opportunity to discuss this recent related finding in our manuscript. We used DTT in our sample workup for MS analysis, and also in wash steps during avidin purification of biotin-tagged proteins (DCP-Bio1 or DYn-2), and therefore dimedone adducts to persulfides would have gone undetected in the present studies. We agree that it

would be of interest to address the potential functional relevance of sulfenylation of pre-existing persulfides for activation, but felt that this would require substantial additional experimentation, which would be complicated by the various technical concerns associated with analysis of persulfides, as well as additional modeling studies that would go beyond the scope of our present manuscript. Nevertheless, we have added a few sentences towards the end of the discussion to address this possibility more directly.

REVIEWERS' COMMENTS:

Reviewer #1 (Remarks to the Author):

The authors have satisfactorily addressed reviewer concerns and the manuscript is much improved.
Minor: the citation for 66-dimedone should go directly after the reagent name, instead of the end of the sentence to avoid confusion.

REVIEWERS' COMMENTS:

Reviewer #1 (Remarks to the Author):

Request: The authors have satisfactorily addressed reviewer concerns and the manuscript is much improved. Minor: the citation for 66-dimedone should go directly after the reagent name, instead of the end of the sentence to avoid confusion.

Response: Reference 40 to d6-dimedone has been moved from the end of the sentence to directly behind the reagent name.